# Structural View of Cryo-Electron Microscopy-Determined ATP-Binding Cassette Transporters in Human Multidrug Resistance

**DOI:** 10.3390/biom14020231

**Published:** 2024-02-17

**Authors:** Wenjie Fan, Kai Shao, Min Luo

**Affiliations:** Department of Biological Sciences, Faculty of Science, National University of Singapore, Singapore 117543, Singapore; e0708160@u.nus.edu (W.F.); shaokai@nus.edu.sg (K.S.)

**Keywords:** multidrug resistance, ABC transporter, cancer, cryo-electron microscopy

## Abstract

ATP-binding cassette (ABC) transporters, acting as cellular “pumps,” facilitate solute translocation through membranes via ATP hydrolysis. Their overexpression is closely tied to multidrug resistance (MDR), a major obstacle in chemotherapy and neurological disorder treatment, hampering drug accumulation and delivery. Extensive research has delved into the intricate interplay between ABC transporter structure, function, and potential inhibition for MDR reversal. Cryo-electron microscopy has been instrumental in unveiling structural details of various MDR-causing ABC transporters, encompassing ABCB1, ABCC1, and ABCG2, as well as the recently revealed ABCC3 and ABCC4 structures. The newly obtained structural insight has deepened our understanding of substrate and drug binding, translocation mechanisms, and inhibitor interactions. Given the growing body of structural information available for human MDR transporters and their associated mechanisms, we believe it is timely to compile a comprehensive review of these transporters and compare their functional mechanisms in the context of multidrug resistance. Therefore, this review primarily focuses on the structural aspects of clinically significant human ABC transporters linked to MDR, with the aim of providing valuable insights to enhance the effectiveness of MDR reversal strategies in clinical therapies.

## 1. Introduction

Metastatic cancer treatment faces a significant challenge, with up to 90% of failures attributed to drug resistance, resulting in poor prognosis and recurrence [1]. Multidrug resistance (MDR) describes the phenomenon where cancer cells exhibit resistance to various chemotherapeutic drugs that are structurally and functionally dissimilar [2,3]. Mechanisms contributing to MDR encompass reduced drug uptake, decreased apoptosis, cytoprotective autophagy, epigenetic regulation, and enhanced DNA damage repair. A major driver of MDR is the increased efflux of cytotoxic drugs by ATP-binding cassette (ABC) transporters [4,5,6].

These transporters, highly expressed in cancer, efflux a wide range of cytotoxic drugs, mainly hydrophobic and amphiphilic molecules like taxanes, vinca alkaloids, anthracyclines, epipodophyllotoxins, and others [7,8,9]. Understanding how these transporters recognize and transport diverse substances is crucial, given their potential as therapeutic targets. Although inhibitors/modulators have been developed, their limited efficacy necessitates the quest for more specific, high-affinity, and low-toxicity inhibitors [10]. Cryo-electron microscopy (cryo-EM) has been instrumental in providing three-dimensional (3D) structural insights into MDR-causing ABC transporters like ABCB1, ABCC1, ABCC3, ABCC4, and ABCG2, shedding light on protein-ligand interactions and opening avenues for effective modulator development to combat drug resistance in cancer [11].

This review focuses on the recent structural advances in human MDR-related ABC transporters, their functional mechanisms in drug resistance, and inhibition strategies from the past decade. It also outlines future research directions, harnessing structural insights to design more potent inhibitors for enhanced drug delivery and MDR reversal in cancer therapy.

## 2. ABC Transporters and Their Roles in MDR

ABC transporters are a diverse group of proteins expressed throughout the body, responsible for actively moving a wide range of substances, both internal and external to cells, across cellular membranes using energy from ATP hydrolysis [12,13,14]. These transporters play crucial roles in maintaining cellular homeostasis and eliminating foreign substances [15]. In humans, there are 48 functional ABC transporters divided into seven subfamilies (ABCA to ABCG), each with distinct structures, functions, substrate preferences, and possibly transport mechanisms as well [13,16]. Their impact on human health and disease, as well as their significance as therapeutic targets, is substantial [11].

Certain members of the ABCB, ABCC, and ABCG subfamilies are implicated in MDR by pumping out cytotoxic chemotherapy drugs from cancer cells, thereby reducing drug concentrations inside the cells [17,18,19]. These transporters also influence the absorption of small molecules in the intestine and transport across the blood–brain barrier, significantly affecting drug pharmacokinetics [20,21]. ABCB1, the first identified ABC transporter, is well-known for its role in drug resistance [22], and subsequent discoveries revealed ABCC1 and ABCG2 as contributors to multidrug resistance in lung and breast cancer cells, respectively [23,24]. The identification and cloning of ABCC1 led to the discovery of eight additional homologs (ABCC2–ABCC6 and ABCC10–ABCC12) known as multidrug resistance-associated proteins (MRPs) [25,26,27].

MDR-causing ABC transporters share a common structural framework, featuring two transmembrane domains (TMDs) embedded in the cell membrane and two nucleotide-binding domains (NBDs) in the cytoplasm responsible for ATP hydrolysis (Figure 1). ABCB and ABCC subfamily members consist of a single polypeptide, or a full transporter, with two TMDs, two NBDs, and a flexible linker connecting them. In contrast, ABCG subfamily MDRs are composed of two half-transporters, each containing one TMD and one NBD, functioning as homo- or hetero-dimers. Each TMD comprises six transmembrane helices. Besides the core TMD–NBD architecture, some MDR-related ABC transporters also possess accessory domains like the N-terminal transmembrane domain 0 (TMD0), containing four or five transmembrane helices, found in certain “long” ABCC subfamilies (Figure 1) [11,28]. The ATP-binding site resides in the NBD’s motor domain, which is highly conserved among all ABC transporters and contains key motifs such as Walker A (P loop), Walker B, the signature motif, A loop, Q loop, D loop, and H loop [11]. ATP hydrolysis takes place at the dimerized NBD interface, adopting a ‘head-to-tail’ orientation, with the conserved motifs playing pivotal roles at this shared interface [29].

## 3. Cryo-EM Study of Drug Recognition, Translocation, and Inhibition of MDR-Related ABC Transporters

The first structure of ABC transporter–vitamin B_12_ importer BtuCD from *E. coli* was resolved in 2002 by crystallography, which set a framework for ABC transporter architecture and mechanism [30]. To date, cryo-EM studies have elucidated the structures of five human ABC transporters associated with multidrug resistance: ABCB1, ABCG2, ABCC1, ABCC3, and ABCC4 [31]. These structural analyses have unveiled crucial insights into these transporters, covering their structural conformation at rest, their mechanism of drug recognition and translocation, as well as details relevant to their inhibition. These findings play a pivotal role in understanding the mechanisms underlying multidrug resistance and offer valuable insights for potential therapeutic interventions aimed at reversing MDR. In the following sections, we will provide a comprehensive summary of the structural information for each of these ABC transporters.

### 3.1. ABCB1

ABCB1, also known as P-glycoprotein (P-gp) and multidrug-resistant protein 1 (MDR1), is an extensively studied 170 kDa protein comprising 1280 amino acids [32]. It was initially characterized in 1976 and later cloned in various cell lines [22,33,34,35]. ABCB1 is widely expressed in normal tissues, including blood–organ barriers, the gastrointestinal tract, kidney, and liver, where it actively participates in the excretion of exogenous substances and xenobiotics [11,36,37,38]. It exhibits poly-specificity, recognizing structurally and chemically unrelated substrates, primarily hydrophobic or weakly amphipathic compounds, and acts as a “hydrophobic vacuum cleaner” to extrude various compounds (Table 1) [39,40]. Elevated ABCB1 expression is associated with MDR in several cancers, including acute myelogenous leukemia, breast cancer, and lung cancer, making it a potential prognostic marker and target for modulators [41,42,43,44]. ABCB1 was initially identified as a key player at the blood–brain barrier in mice. Its deletion resulted in elevated drug levels and reduced drug elimination in various tissues, particularly within the central nervous system [38]. The expression level of ABCB1 decreases with the natural aging process in brain endothelial cells, leading to the accumulation of amyloid beta in the brain [45].

#### 3.1.1. Drug Binding and Translocation of ABCB1

ABCB1 is a full transporter with canonical structures featuring two transmembrane domains (TMDs) and two nucleotide-binding domains (NBDs). The first crystal structures of ABCB1 from mice were determined in 2009, unveiling the inward-facing conformations of both apo and cyclic peptide inhibitor-bound structures (Figure 2A) [39]. The structure of apo-ABCB1 at 3.8 Å reveals an internal cavity of approximately 6000 Å^3^, with a 30 Å separation between the two nucleotide-binding domains (NBDs). In two additional ABCB1 structures with cyclic peptide inhibitors, distinct drug binding sites within the internal cavity showcase stereo-selectivity determined by hydrophobic and aromatic interactions. These structures of ABCB1 unveil a molecular foundation for poly-specific drug binding [57]. In another study, a nanobody Nb592 is bound to the C-terminal side of the first nucleotide-binding domain. This nanobody effectively inhibits the ATP hydrolysis activity of mouse P-glycoprotein by impeding the formation of a dimeric complex between the ATP-binding domains, a crucial step for nucleotide hydrolysis. Recent cryo-EM studies have provided insights into the structure of human ABCB1 (hABCB1) or human-mouse chimeric ABCB1 (hmABCB1) in complex with various ligands/ATP (Figure 2B). In particular, the use of inhibitory monoclonal antibodies UIC2 and MRK16 has facilitated higher-resolution determinations [46,47,58]. These antibodies target the extracellular loops of ABCB1 on the cell membrane’s external side [59]. The complex structures of ABCB1 with chemotherapy drugs Taxol (PDB code: 6QEX) and vincristine (PDB code: 7A69) in occluded conformations, bound to UIC2 and MRK16, respectively, illustrate the poly-specific binding mode of ABCB1 [47,58]. The substrate-binding pocket is situated at the center of two TMDs within the cell membrane, characterized by a flexible and aromatic central cavity enriched with hydrophobic and aromatic residues (Figure 2C). A single drug molecule binds within this enclosed central cavity and interacts with surrounding transmembrane helices (TMs). TM4 and TM10 undergo ordered kinks toward the pseudo-symmetric axis, forming a gate region that accommodates substrate molecules (Figure 2D). Notably, the central pocket can accommodate drug molecules in multiple orientations due to the large cavity size and plasticity provided by aromatic residues that could mediate multiple interactions with various molecules [47,58].

Upon ATP binding, hABCB1 adopts an outward-facing conformation (PDB code: 6C0V), with the two NBDs forming a closed dimer and binding two ATP molecules. TM4 and TM10 adopt continuous helical structures in the outward-facing state, essential for closing the intracellular gate (Figure 2D) [36]. This conformation results in the reorientation and opening of the central binding cavity towards the extracellular side, facilitating substrate release during the transport cycle [36].

#### 3.1.2. Inhibition of ABCB1

Since the discovery of its multidrug efflux function, ABCB1 has remained a key focus in cancer therapy. The first structure of mouse ABCB1 guided the development of inhibitory molecules [39]. Three generations of inhibitors/modulators for ABCB1 have been developed, aiming to overcome issues such as low affinity, high toxicity, complex drug-drug interactions, and unpredictable pharmacokinetic interactions [60,61,62]. These inhibitors, while showing promise in overcoming MDR, have faced challenges in clinical trials [63,64,65].

Structural studies have enhanced our understanding of the inhibition mechanisms of ABCB1. The crystal structures of apo and cyclic peptide-inhibited ABCB1 reveal the inward-facing confirmation, signifying an initial stage in the transport cycle that is conducive to drug binding. Substrates or inhibitors may exist during their initial entry into the internal cavity, potentially elucidating their role in regulating ATPase activity. Determinations of UIC2-bound or MRK16-bound ABCB1 with inhibitor zosuquidar (PDB codes: 6FN1, 6QEE, and 7A6F), elacridar (PDB code: 7A6C), tariquidar (PDB code: 7A6E), and encequidar (PDB code: 7O9W) have provided insights into their occluded conformations (Figure 2E) [46,47,58,66]. These structures reveal a pair of inhibitor molecules occupying the central cavity, where substrate drug molecules occupy. Several residues comprising the central cavity, including Tyr310, Phe343, Gln725, Gln949, Tyr953, and Phe983, both participate in substrate drug molecules and inhibitor molecules binding (Table 1). Notably, one of the two inhibitor molecules extends into the “vestibule” and/or “access tunnel” facing the cytoplasmic gate [58]. In most cases, one U-shaped molecule binds to the central pocket, while one L-shaped molecule extends into the vestibule/access tunnel (Figure 2F,G). TM9 plays a role in distinguishing substrates and inhibitors by moving into the vestibule and access tunnel after substrate binding, acting as an “initiator” for extrusion release. Additionally, the molecule in the access tunnel may function as a noncompetitive inhibitor [58].

New strategies such as genetic modifications were carried out to study the roles of ABCB1 transmembrane helixes (such as TM1/7 and TM6/12) and discuss the potential DNA therapies to change substrate specificity and transport direction to overcome MDR [67,68]. Substituting twelve residues on TM1 and TM7 with alanine did not affect the expression level or alter the conformation, as confirmed by ABCB1-specific antibodies. Intriguingly, the ABCB1 variant, resulting from these substitutions, exhibited a loss in its capacity to transport a wide range of substrates and displayed a reduced basal ATPase activity [67]. In another separate study, 14 residues were mutated on both TM6 and TM12, rendering the protein incapable of effectively extruding the majority of tested substrates from cancer cells. Astonishingly, this variant exhibited a novel function, demonstrating its ability to import four distinct substrates [68]. Such findings represent a groundbreaking paradigm shift in the ongoing endeavors to overcome drug resistance.

### 3.2. ABCG2

ABCG2, also known as breast cancer resistance protein (BCRP), is a 72 kDa protein comprising 655 amino acids and was first identified in 1998 [23]. It is widely distributed in various tissues and tissue barriers, including the blood–brain barrier, blood–testis barrier, placenta, liver, kidney, small intestine, and mammary glands [48,69]. ABCG2 plays a pivotal role in the transport of sterol sulfate and urate, with deficiency of ABCG2 function being a major contributor to hyperuricemia and gout [70]. This transporter recognizes and transports a broad spectrum of substrates, primarily hydrophobic, polycyclic, and relatively flat molecules, encompassing endogenous substrates such as sulfate conjugates of steroids, uric acid, and porphyrins, as well as exogenous cytotoxic compounds (Table 1) [51,71]. Moreover, ABCG2 has been implicated in conferring resistance to various chemotherapy drugs, including topotecan, SN-38, mitoxantrone, doxorubicin, and several tyrosine kinase inhibitors (TKIs) [71,72], contributing to multidrug resistance in cancer, as observed in colon cancer [73]. ABCG2 is situated on the luminal side of brain endothelial cells, serving as a barrier to the entry of xenobiotics/drugs such as methotrexate, mitoxantrone, and topotecan, as well as endogenous metabolites into the brain [21]. ABCG2 is also upregulated in the Alzheimer’s brain with cerebral amyloid angiopathy, potentially functioning as a gatekeeper to impede the entry of circulatory Aβ into the brain [74].

#### 3.2.1. Drug Binding and Translocation of ABCG2

ABCG2 is a half transporter consisting of one NBD at the N-terminal and one TMD at the C-terminal, forming homodimers. Multiple structures of human ABCG2, both with and without ligands or ATP, have been elucidated in various conformations (Figure 3A), main conformations including apo-closed (Figure 3B), inward-facing (Figure 3C) and outward-facing (Figure 3D) states. In the absence of ligands and nucleotides, ABCG2 adopts an apo-closed conformation as the resting state, characterized by collapsed TM helices and separated NBDs (PDB code: 6VXF) (Figure 3B) [50]. In complex with two Fab fragments of the inhibitory conformational antibody 5D3 (PDB code: 5NJ3), which recognizes the extracellular loops of ABCG2 at an angle of approximately 35 degrees to the cell membrane, ABCG2 is observed in an inward-facing conformation [75]. These Fab fragments are utilized in subsequent structural studies to facilitate high-resolution determinations without compromising ATP hydrolysis or transport function. The allosteric inhibition by 5D3-Fab is attributed to clamping ABCG2 and preventing it from adopting an outward-facing conformation [75]. Human ABCG2 bound with various substrates has been extensively studied in inward-facing conformations or turnover states, with or without 5D3 Fab. These substrates include the endogenous substrate E_1_S (PDB codes: 6HCO and 7OJ8), chemotherapy drugs such as mitoxantrone (PDB codes: 6VXI and 7NFD), SN38 (PDB code: 6VXJ), tariquidar (PDB codes: 7NEQ, 8BHT, and 8BI0), and topotecan (PDB codes: 7NEZ, 7OJH, and 7OJI) [49,50,51,76,77]. These structures provide valuable insights into conformational changes during drug translocation.

The structures of ABCG2 reveal a more compact organization and shorter transmembrane domains (TMDs) compared to B subfamily members. Two notable features are the presence of a deep, slit-like, and hydrophobic cavity 1 within the central cavity of the two TMDs and a smaller cavity 2 (external cavity) located below external loop EL3, separated by a “leucine plug.” These features are closed in the inward-facing state (Figure 3C) [75]. This structural arrangement explains ABCG2’s preference for flat substrates, in contrast to ABCB1, which favors globular hydrophobic substrates. The structures reveal the accommodation of a single substrate molecule in the slit-like cavity 1, formed by TM2 and TM5 below the leucine plug. The precise location and orientation of substrates shed light on the importance of specific residues, such as Phe439 in Π-stacking interactions and Asn436 in hydrogen bond formation (Figure 3E). The Arg482 has long been recognized as a ‘hot spot’ site, and its mutation affects the substrate specificity of ABCG2 [78,79]. Positioned on TM3 outside of slit-like cavity 1, Arg482 does not directly contact substrates within cavity 1. Instead, it interacts with TM2, which houses the crucial residue Phe439 [77]. The mutation of Arg482 to glycine or threonine, for example, would reshape the slit-like cavity 1 and thus alter substrate affinity and specificity. The substitution of Gln141 with lysine, known to induce hyperuricemia and gout [80], is located at the nucleotide-binding domain but close to the TMD–NBD interface. The mutation Q141K might disrupt the previous interaction network, influencing the coupling between TMD and NBD and resulting in ABCG2 dysfunction. Additional mutations, such as R147W, F373C, R383C, and S476P, are all distributed at the interface of TMD and NBD. These mutations could change the coupling dynamics and transport activity [81]. Certain mutations like M71V, T153M, and T434M also lead to functional defects of ABCG2 [81]. Their residues are spread over protein internal space, and their mutations may cause significant disruptions of protein structure, rendering the transporter inactive.

ABCG2 has also been observed in an ATP-bound NBD-closed conformation, representing a post-translocation state, characterized by a collapsed substrate-binding cavity 1 and an open cavity 2 facing outward (PDB code: 6HBU) (Figure 3D) [49]. Further insights into turnover states have been gained in complexes with different substrates/drugs (PDB codes: 7OJ8, 7OJH, 7OJI, 8BHT, and 8BI0), which represent transitional states between the inward-facing and outward-facing conformations and reveal how ATP binding contributes to the closure of the cytoplasmic side of the TMDs [76,77]. However, the fully outward-facing state with the open leucine gate for substrate release has not been observed, likely due to its transient nature.

#### 3.2.2. Inhibition of ABCG2

The inhibition of ABCG2 by various molecules has been extensively investigated. The first ABCG2 inhibitor, FTC, was ruled out due to its neurotoxicity [82]. Its derivative, Ko143, demonstrated lower toxicity and increased efficiency but lacked selectivity for ABCG2 [83,84]. Several inhibitors, including tyrosine kinase inhibitors (TKIs) and anti-HIV drugs, have been explored, but none have progressed to clinical use due to safety and efficacy concerns [71].

The inhibition of ABCG2 has been primarily studied using derived inhibitor molecules and inhibitory antibodies (Figure 3F). Structures of ABCG2 bound to the Ko143-derived inhibitor molecule MZ29 (PDB codes: 6FFC and 6ETI) and the tariquidar-derived inhibitor molecule MB136 (PDB code: 6FEQ) have been determined [48]. Depending on their shape, two MZ29 molecules or one MB136 molecule can symmetrically or asymmetrically occupy cavity 1 (Figure 3G). Additionally, imatinib has been suggested as a potential inhibitor similar to Ko143, with one imatinib molecule spanning the top of cavity 1, acting as a wedge to stabilize the inward-facing state (PDB code: 6VXH) [50].

Recently, inhibitor nanobodies Nb8 (PDB code: 8P7W), Nb17 (PDB code: 8P8A), and Nb96 (PDB code: 8P8J) have been discovered as an alternative mode of inhibiting ABCG2 by binding to NBDs [85]. These small nanobodies (13–14 kDa) reduce ATPase activity to an extent and largely abolish substrate transport. The structures reveal various binding epitopes of three nanobodies on the cytosolic-side NBDs while all allosterically lock the transporter in the inward-open conformation. Two copies of Nb8 bind at the apex of two NBDs, while a single copy of Nb17 binds and interacts with both NBDs and a single Nb96 binds at one NBD (Figure 3H). They are proposed to stabilize certain residues in the NBDs to interfere with ATP binding and hydrolysis and limit conformational changes. Therefore, they interfere with ATP hydrolysis and substrate transport and prevent the transition to the outward-facing conformation [85].

### 3.3. ABCC Family: ABCC1, ABCC3, and ABCC4

Nine members of the ABCC subfamily, specifically ABCC1–6 and ABCC10–12, are known as multidrug resistance-associated proteins (MRPs) and are denoted as MRP1–9 [86]. These proteins play a significant role in expelling cytotoxic anticancer drugs, contributing to multidrug resistance (MDR). Additionally, their presence at the blood–brain barrier (BBB) presents challenges in the effective treatment of neurological disorders [87]. Multidrug resistance transporters expressed in brain parenchyma may facilitate the overall export of xenobiotics from the central nervous system, which will also hand neurological treatment drugs off to the barrier tissues. ABCC1 is involved in the extraction of E_2_17βG at the BBB and provides a barrier function by extruding conjugated metabolites [88]. Additionally, ABCC1 was found to play a remarkable role in cerebral Aβ clearance and accumulation [89]. While they share functions related to MDR, each MRP within the ABCC subfamily also has distinct functions [90].

Structurally, MRPs in the ABCC subfamily exhibit common features. They all possess the conserved interfacial lasso motif, which plays a crucial role in protein trafficking and function. This motif was initially observed in the structure of the cystic fibrosis transmembrane conductance regulator (CFTR), also known as ABCC7 [91,92]. Approximately half of the MRPs (MRP1, 2, 3, 6, 7) contain an N-terminal transmembrane domain (TMD0), which is known to mediate interactions between the transporter and other proteins [93,94]. They feature one consensus ATPase site at NBD2 and one degenerate ATPase site at NBD1, with only the consensus site capable of ATP hydrolysis [27,95].

Among these MRPs, the structures of ABCC1, human ABCC3, and human ABCC4 have been elucidated, both independently and in complex with substrates or inhibitors. These structural insights provide valuable information regarding the mechanisms of drug transport and inhibition related to multidrug resistance (MDR).

#### 3.3.1. ABCC1

ABCC1, also known as MRP1, is a substantial 190 kDa (1531 amino acids) protein initially characterized in 1992 [24]. It is typically localized on the basolateral membrane of polarized epithelial cells and exhibits expression in various tissues, with notably higher levels in the kidney, lung, testes, skin, placenta, and the blood–brain barriers [96,97]. ABCC1 plays a versatile role in recognizing a wide range of substrates, including organic anion conjugates of both endogenous and xenobiotic compounds (Table 1) [90]. Its substrate repertoire encompasses hydrophobic and hydrophilic drugs, such as vinca alkaloids, anthracyclines, and epipodophyllotoxins [97]. The involvement of ABCC1 in drug resistance has been documented in various cancers, including acute myeloblastic and lymphoblastic leukemia, prostate cancer, non-small-cell lung cancer, neuroblastoma, and breast carcinoma [63,98,99].

##### Drug Binding and Translocation of ABCC1

Structurally, ABCC1 consists of a single polypeptide containing three TMDs, comprising two canonical TMDs and an N-terminal TMD0, collectively housing 17 transmembrane helices. To date, only structures of bovine ABCC1, which shares 91% sequence identity with human ABCC1, have been determined, serving as valuable templates for studying the human homolog. The apo form of ABCC1 (PDB code: 5UJ9) has been resolved in an inward-facing conformation [52]. Additionally, the structure of ABCC1 bound to its physiological substrate, leukotriene C4 (LTC_4_) (PDB code: 5UJA), was also determined to be in an inward-facing conformation but with a closer distance between the two nucleotide-binding domains (NBDs) (Figure 4A) [52]. This structure elucidates the direct recruitment of the substrate from the cytosol and reveals the bipartite binding site, which comprises partially positively charged (the P-pocket) and partially hydrophobic (the H-pocket) regions to accommodate hydrophobic moieties (Figure 4B). LTC_4_ binds within the central cavity formed by the two TMDs, establishing an intricate interaction network with both TMD bundles (Figure 4C,D). LTC_4_ binding at the center of the two TMDs brings the two NBDs approximately 12 Å closer, explaining the increased ATPase activity observed in the presence of LTC_4_ and the concomitant rearrangement of residues within the binding site to adapt to the substrate (Figure 4C).

Structural studies have also provided insights into the ATP-bound pre-hydrolytic state of ABCC1, demonstrating an outward-facing conformation (PDB code: 6BHU), as well as the ATP- and ADP-bound post-hydrolytic turnover state (PDB code: 6UY0) [100,101]. ATP binding leads to the opening of the transport pathway and conformational rearrangements at the binding site, facilitating substrate release prior to ATP hydrolysis, which subsequently resets the transporter to its resting state following the rate-limiting step of NBD dimer dissociation [100,101].

##### Inhibition of ABCC1

In terms of inhibition, ABCC1-specific inhibitors are less abundant compared to those for ABCB1 and ABCG2. Notably, the inhibition of ABCC1-mediated drug efflux has been achieved using ABCB1 and ABCG2 inhibitors [102]. Some agents, such as indomethacin, verapamil, and its derivatives, have shown sensitivity to ABCC1 [103,104]. However, most ABCC1 inhibitors exhibit off-target effects due to their low specificity [53,105]. To address this issue, macrocyclic peptides have emerged as promising inhibitors, offering a larger binding interface, increased affinity, and higher selectivity. The structure of ABCC1 in a complex with a macrocyclic peptide inhibitor known as CPI1 has been determined in an inward-facing conformation (PDB code: 8F4B) [53]. The inhibitor molecule competitively binds to the same central pocket as the substrate, occupying nearly the entire transmembrane pathway (Figure 4C,E).

##### GSH and GSH-Conjugated Molecule Transport of ABCC1

GSH plays a pivotal role in the transport activity of certain ABCCs, notably ABCC1 [63]. ABCC1 demonstrates the ability to efflux various molecules either in the presence of GSH or in its GSH-conjugated form [106]. This transporter facilitates the cellular efflux of a diverse range of organic anions, xenobiotics, and their conjugated metabolites [63,107]. Many of these organic anions, such as LTC_4_ and E_2_17βG, are conjugated to GSH. Various xenobiotics, including verapamil, vincristine, and indinavir, can enhance ABCC1’s ability to transport GSH, and certain ABCC1 modulators have been found to facilitate their binding and inhibitory activity by GSH [108].

The structure of GSH conjugated molecules bound to ABCC1, such as LTC_4_, has been elucidated [52], revealing the molecular mechanism of GSH accumulated transport LTC_4_ by ABCC1. LTC_4_ binds to ABCC1 within the transmembrane pathway between the transmembrane bundles, with the binding site divided into two parts: a hydrophobic area (H-pocket) encompassing the LTC_4_ lipid tail and a positively charged pocket (P-pocket) coordinates the GSH moiety. This binding coordination likely optimizes the substrate fit within relatively separate hydrophobic and positively charged pockets. The recently determined cryo-EM structure of ABCB6 bound to GSH and GSH-conjugated hemin provides an additional perspective for understanding these processes [109]. GSH emerges as a key regulator of the binding of metal porphyrins, acting as an anchor for the positively charged metal ions. The binding of metal porphyrin, coupled with GSH, stabilizes a conformation where the two NBDs are closer, facilitating ATP binding and hydrolysis. Unexpectedly, two GSH molecules are found bound outside the central cavity, a region where a detergent molecule occupies in a bacterial homolog of ABCB6, suggesting a non-essential role for protein function [109,110].

#### 3.3.2. ABCC3

ABCC3, also known as MRP3, is a substantial 169 kDa (1527 amino acids) protein that was initially identified in 1997 [26]. It primarily localizes to the basolateral membrane of polarized cells and exhibits widespread expression in tissues such as the liver and kidney [111]. This transporter specializes in the translocation of organic anionic conjugated compounds and can transport a diverse array of endogenous metabolites, including bilirubin glucuronides, bile acids, and steroid hormones (Table 1). It serves essential roles in maintaining the homeostasis of steroid hormones and mediating the efflux of accumulated bile acids in the liver [54,111]. As a member of the MRP family, it has been reported to confer resistance to a limited number of anticancer drugs, including etoposide, teniposide, methotrexate, and vincristine [112,113]. Elevated expression levels of ABCC3 have been observed in hepatocellular carcinoma and non-small-cell lung cancer, where it is suggested to serve as a marker for multidrug resistance (MDR) and a predictor for poor clinical outcomes [114,115].

Similar to ABCC1, ABCC3 is a full transporter consisting of three transmembrane domains (TMDs) and two nucleotide-binding domains (NBDs), featuring both a consensus and a degenerated ATPase site. The apo form of human ABCC3 (PDB code: 8HVH) adopts an inward-facing conformation [54]. Furthermore, the structure of ABCC3 in complex with two conjugated hormone substrates, E_2_17βG (PDB code: 8HW2) and DHEAS (PDB code: 8HW4), has also been determined, revealing inward-facing conformations (Figure 5A) [54]. Analysis of these three ABCC3 structures indicates minimal conformational changes upon substrate binding, except for side-chain rotations of key binding pocket residues. The substrate-binding pocket within the central cavity exhibits a V-shaped configuration capable of accommodating two substrate molecules, notably E_2_17βG, in an asymmetric arrangement (Figure 5B,C). This binding pocket comprises a substantial hydrophobic surface and two smaller polar patches. Complementing biochemical studies, it is evident that polar interactions play a pivotal role in substrate binding, with particular significance attributed to Arg1193 and Arg1245, which form salt bridges with one of the two bound molecules.

#### 3.3.3. ABCC4

ABCC4, initially identified in 1997 alongside ABCC3 and ABCC5, is a substantial 150 kDa (1325 amino acids) protein [26]. It is found in both apical and basolateral membranes and exhibits widespread expression in nearly all tissues [116]. ABCC4 serves as a transporter for a wide array of physiological substrates, including cyclic nucleotides, steroid conjugates, folate, and prostaglandins (PGE1 and PGE2) (Table 1) [117]. Additionally, it plays a crucial role in transporting xenobiotics such as antibiotics, antiviral agents, and anticancer drugs [118]. Elevated expression levels of ABCC4 have been identified in neuroblastoma, prostate cancer, hepatocellular carcinoma, breast carcinoma, and non-small-cell lung cancer [98,119,120,121,122]. ABCC4 has been implicated in conferring resistance to a broad range of clinical antineoplastic drugs, including thiopurines, antifolates, and camptothecins [55].

As a “short” member of the MRP family, ABCC4 comprises two transmembrane domains (TMDs) and two nucleotide-binding domains (NBDs) without an additional TMD0. Notably, it features a C-terminal PDZ-binding motif, setting it apart from other MRPs [55]. The apo form of human ABCC4 (PDB codes: 8I4B, 8BJF, and 8IZ8) adopts an inward-facing conformation, representing the resting state, akin to ABCC1 and ABCC3 [55,56,123]. Structures of ABCC4 bound with substrates or substrate analogs, including aspirin (PDB code: 8J3W), U46619 (PDB code: 8I4C), PGE1 (PDB code: 8IZ9), and PGE2 (PDB code: 8BWR), as well as chemotherapy drugs methotrexate (PDB code: 8BWP) and topotecan (PDB code: 8BWQ), have all been observed in inward-facing conformations similar to the apo form (Figure 6A) [55,56,123]. Additionally, ABCC4 bound with U46619 and ATP (PDB code: 8J3Z) reveals an outward-facing occluded conformation, likely representing a transient state just before substrate release [123]. Substrates bind with ABCC4 in the central pocket primarily through extensive hydrophobic interactions, complemented by electrostatic interactions (Figure 6B,C). Several key residues within the binding pocket have been identified as crucial for drug efflux [55].

The ATP-bound state of ABCC4 is observed in two distinct forms, with or without ligands (PDB codes: 8BWO and 8J3Z), both adopting outward-facing occluded conformations. In this conformation, the two NBDs are in a closed configuration, and the central cavity is sealed both to the intracellular and extracellular sides [55,123]. An ATP-bound ABCC4 structure (PDB code: 8IZA) in an outward-facing conformation, resembling an open state toward the extracellular side, has also been reported [56].

Besides substrates or drugs, inhibitors such as dipyridamole (PDB code: 8I4A) and sulindac (PDB code: 8IZ7) are also observed to bind in the central binding pocket in highly similar inward-facing conformations (Figure 6D,E), but they establish more interactions with transmembrane domains compared to substrates [56,123].

## 4. The Mechanisms of Drug Recognition, Translocation, and Inhibition in Multidrug Resistance

### 4.1. Putative Mechanism of Multidrug Recognition

Most MDR-causing ABC transporters with determined structures feature a central substrate-binding pocket characterized by remarkable flexibility and adaptability to accommodate a wide range of substrates. For ABCB1, its expansive globular hydrophobic pocket is rich in aromatic residues, making it suitable for binding generally hydrophobic molecules. In contrast, ABCC1, ABCC3, and ABCC4 employ large bipartite pockets to recognize amphiphilic substrates. ABCG2 shows a preference for flat molecules, which can be revealed by the compact central cavity comprising a deep, slit-like cavity 1.

However, the overall cavity is predominantly lined with hydrophobic and aromatic residues, providing the necessary versatility to engage in various interactions with substrates, including pi–pi, pi–cation, pi–anion, hydrophobic interactions, and more. Additionally, the spacious pocket accommodates substrates of varying sizes. Together, the ample space, flexible residues, and aromatic components framing the pocket collectively contribute to the multidrug recognition capability exhibited by these MDR-causing ABC transporters, facilitating multidrug resistance.

### 4.2. Mechanism of Drug Translocation

The mechanism of drug translocation by MDR-related ABC transporters can be elucidated through a general and straightforward “alternating access” model, necessitating significant conformational changes between two primary states: inward-facing and outward-facing [124]. This transition involves the binding of ATP and substrates, resulting in a shift to the outward-facing conformation, during which substrates are expelled to the extracellular side. Subsequent ATP hydrolysis resets the transporter to the inward-facing conformation, passing through intermediate states (Figure 7) [125].

In the absence of ATP and ligands, both ABCB1 and members of the ABCC subfamily adopt an inward-facing conformation as their default state. Substrate binding triggers conformational changes, for example, bringing the two NBDs into closer proximity and preparing them for ATP binding and hydrolysis. In the case of ABCB1, the pocket is sealed from the cytosolic side by TM4 and TM10, while for ABCC1, substrate binding induces the rearrangement of pocket residue side chains. However, the structures of ABCC3 and ABCC4 reveal no significant NBD movement after substrate binding; instead, conformational changes are confined to the central binding pocket.

Upon ATP binding, the two halves of the transporter shift closer together, causing the substrate pocket to collapse. This transition from an inward-facing to an outward-facing conformation facilitates substrate release before ATP hydrolysis. The reshaping of the pocket reduces substrate affinity and stiffens the TMs in the outer leaflet, facilitating substrate release. Importantly, ATP hydrolysis resets the transporter to its resting inward-facing state [36]. Notably, NBD dissociation occurs more slowly than ATP hydrolysis [100]. Moreover, the outward-facing occluded conformation of substrate-bound ABCC4 (PDB code: 8J3Z) hints at a potential transient state preceding substrate release [123].

ABCG2 follows a similar transport mechanism but starts in a closed conformation during its resting state, featuring collapsed TMDs and open NBDs. Substrate binding triggers the opening of the TMDs into an inward-facing conformation. Prior to reaching the inward-facing state, the transporter may undergo turnover states 1 or 2, depending on the specific substrates involved [76,77].

Together, while the prevailing model for MDR-causing ABCs involves an ‘alternating access’ mechanism, minor distinctions exist among them. In the initial phase of drug recognition, most MDR-causing ABCs bind the drug within a central pocket while adopting an inward-open conformation. In contrast, ABCB1 features sealing by the two TM helices, TM4 and TM10, effectively preventing the bound drug from returning to the cytosol. Moreover, for drug release, certain MDR-causing ABCs assume a wider outward-opening conformation, as seen in the case of ABCB1, while others adopt a more compacted structure with a narrower opening towards the exterior. These variations provide more dynamics in the translocation mechanism and likely contribute to the specificity of MDR-causing ABCs in terms of substrate selection and drug efflux efficiency.

### 4.3. Inhibition Mechanisms of MDR-Related ABC Transporters

The inhibition mechanisms elucidated by high-resolution three-dimensional structures can be categorized into two main approaches: one involves relatively small-size molecules or peptides, while the other employs inhibiting antibodies. The higher binding affinity of small-molecule inhibitors is evident through their extensive interactions and stronger contacts with the transporter, surpassing those of substrates. This can be observed in two ways: either through a single large inhibitor molecule with a substantial contact interface or by the binding of two smaller molecules within the binding pocket. Conversely, inhibiting antibodies primarily function by stabilizing the transporter in a specific conformation. Nanobodies and antibodies offer several advantages over small-size molecules, such as low toxicity, high specificity, and absence of off-target effects, albeit at a higher cost. Nanobodies, in particular, exhibit enhanced stability, improved solubility, and swift, targeted distribution compared to antibodies. On the other hand, small molecule inhibitors are generally more affordable and easier to administer but often come with the drawback of potential off-target effects [126].

In the case of inhibitor-bound ABCB1, the paired inhibitors exhibit stronger interactions with the binding cavity compared to chemotherapy drugs. The binding mode of inhibitor molecules is thought to sterically impede the current occluded conformation, preventing the conformational changes necessary for transitioning to the outward-facing state, thereby inhibiting drug efflux [47,58]. The antibodies exert their inhibitory effect on ABCB1 by binding to external loops, effectively clamping these loops together. This binding prevents ABCB1 from transitioning into the outward-facing conformation [47,58]. As a result, the transporters are inhibited in an occluded conformation. It is noteworthy that the crystal structure of nanobody-inhibited ABCB1 (PDB code: 4KSD) was published in 2013 [57]. In this structure, nanobody Nb592 binds to the C-terminal side of the first NBD, robustly inhibiting the ATP hydrolysis activity of mouse ABCB1 by impeding NBD dimerization. This action locks ABCB1 in an inward-facing conformation.

For ABCG2, inhibitors and substrates share overlapping binding contacts with surrounding residues within the same binding pocket. However, inhibitors can occupy a larger space in the cavity, either in the form of a single large molecule or two smaller molecules, resulting in more potent interactions with ABCG2 [48]. Much like ABCB1, the ABCG2 antibody also engages with extracellular loops of ABCG2 [75]. Interestingly, two copies of the antibody symmetrically bind to the ABCG2 homodimer. However, it is notable that a single antibody is sufficient to effectively inhibit the ATPase activity of ABCG2. This binding mechanism hinders the active transport of ABCG2 by clamping the ABCG2 monomers together, effectively immobilizing the transporter in an inward-facing conformation. Additionally, newly developed nanobodies are demonstrated to allosterically bind to ABCG2 NBDs in various modes on the cytosolic side. These inhibitory binders lock the NBDs and trap the transporter in the inward-open state, preventing it from transitioning to the outward-facing conformation and inhibiting ATP hydrolysis [85]. In the case of ABCB1, the nanobody is positioned centrally between the two separated NBDs, obstructing ATP hydrolysis and rendering NBD dimerization unattainable. Conversely, with ABCG2, three distinct nanobodies all bind to the outside of the NBD, where the two NBDs are typically in contact with each other. Despite this, these nanobodies effectively constrain ABCG2 in an inward-open conformation, thereby blocking the transporter’s transport activity. These insights may pave the way for the development of inhibitors with novel binding modes beyond the substrate-binding pocket.

In the cases of inhibitor-bound ABCC1 and ABCC4 in the ABCC subfamilies, extensive interactions and larger spatial occupancy are observed in comparison to substrates, explaining their higher affinity. Unlike LTC_4_, which induces the closure of two NBDs in ABCC1, CPI1 traps ABCC1 in a conformation resembling the apo form. This action is presumed to prevent conformational changes and the rearrangement of residues required for substrate transport [53].

In summary, inhibitors have been shown to bind to central cavities to hinder substrate binding or to NBDs to impede conformational changes. The binding of inhibitors can effectively trap the transporters in an inward-facing apo state (as seen in ABCC1 and ABCG2) or stabilize an occluded state with a closed cytoplasmic entrance, preventing the transition to the outward-facing state (as observed in ABCB1). These structural insights shed light on the intricate inhibition mechanisms and are poised to accelerate future drug discovery and design efforts.

## 5. The Unknowns Pending Future Studies

### 5.1. Central Linker with Unknown Structures and Functions

In the context of ABC transporters, particularly full transporters like ABCB1 and members of the ABCC subfamily, a fundamental aspect is the need for these transporters to coordinate the actions of their two halves, namely the transmembrane domains (TMDs) and the nucleotide-binding domains (NBDs). Unlike half transporters like ABCG2, where each half functions independently, the TMDs and NBDs in full transporters are encoded within a single polypeptide [127]. To ensure proper coordination between these halves, a long and flexible linker is essential, bridging the first NBD and the second TMD. These linkers typically consist of 50–100 amino acid residues and are characterized by their high charge, disorder, and flexibility, and the sequence similarity is quite low among different ABCs (Figure 8) [128,129]. Despite the availability of numerous structural data, the architecture and functions of these central linkers remain enigmatic.

In the case of ABCB1, it has been reported that the central linker strongly interacts with tubulins within cells [130]. Manipulation of the ABCB1 central linker suggests its involvement in coordinating the functions of the two halves by facilitating interactions between the two ATPase sites and potentially influencing substrate specificity [129,131]. Shortening this linker by 34 residues has been proposed to limit its conformational flexibility, affecting substrate transport [132]. Similarly, the ABCC1 linker has been shown to interact with tubulins [133]. Additionally, the linker has been found to bind to the ATP synthase α subunit, a binding that can be enhanced by phosphorylation, suggesting a potential role in regulating ATP levels [134].

It has also been suggested that domain fusion via linkers may represent an evolutionary adaptation of the protein, with linker length potentially correlated with thermodynamics [135]. The central linker is evidently crucial in the translocation cycles of these transporters and is of great significance in the study of transport mechanisms. Nevertheless, the precise functions of the central linker remain elusive, warranting further investigation in future research.

### 5.2. TMD0 with Less Clear Functions in the Context of ABC Transporters

The N-terminal additional transmembrane domain, TMD0, is a distinctive feature found exclusively in the ABCB and ABCC subfamilies. It typically comprises approximately 200 amino acids and forms a rigid bundle of four or five transmembrane helices attached to the core domain of the transporter. TMD0 is typically linked with TMD1 by a flexible linker with missing electron density [54,101,136]. Despite the elucidation of several TMD0 structures, their functions remain largely unclear.

TMD0 has been reported to mediate protein-protein interactions and recruit interacting proteins. For example, ABCC8 (SUR1) TMD0 is essential for normal KATP channel function by interacting with the potassium channel Kir6.2, and the TMD0s of ABCB2/3 (TAP1/2) are involved in the recruitment of tapasin for peptide-loading complex formation [93,94]. Some studies have also suggested a role for TMD0 in subcellular targeting, including lysosomal trafficking of ABCB6 and ABCB9 (TAPL) and apical localization of ABCC2 [137,138,139].

Current research indicates that TMD0 may not directly influence the transport of substrates. For example, ABCC1 without TMD0 can still transport its endogenous substrate LTC_4_ and exhibit comparable ATPase activity to the wildtype [52,140]. Truncation of TMD0 in ABCC3 shows similar ATPase activity and substrate transport compared to full-length wildtype [54]. Although TMD0 appears to be dispensable for substrate binding, questions remain regarding whether it may play a role in substrate transport during the transporter’s functional cycles and its potential contribution to the occurrence of multidrug resistance (MDR). Further research is needed to address these questions and elucidate the full range of TMD0 functions in ABC transporters.

### 5.3. Structure-Based Drug Design

The quest to overcome multidrug resistance (MDR) in cancer patients has spurred extensive efforts in drug design targeting MDR-causing ABC transporters. While numerous inhibitors have been developed, none has demonstrated significant efficacy in reversing MDR in human clinical trials, often due to issues of low specificity, affinity, or undesirable side effects [141]. However, the growing availability of high-resolution structures of human MDR-related ABC transporters has opened up new possibilities for structure-based inhibitor design.

High-resolution cryo-EM structures of MDR-related ABC transporters bound to ligands provide valuable insights into the interactions between proteins and ligands. This wealth of structural information facilitates the development of effective inhibitors, whether by screening for new potential compounds or by optimizing existing inhibitors for enhanced properties. Notably, all investigated small molecules have been found to bind within the central cavity formed by the two transmembrane domains (TMDs). Key residues involved in interactions within the binding pocket, such as Phe439 in ABCG2 for Π-stacking interactions, can serve as the foundation for designing pharmacophores that target these binding pockets [142].

The distinct properties of these binding pockets correspond to the substrate specificities of the transporters, each with its own preference for different substrates. For instance, ABCB1 features a globular hydrophobic pocket, while ABCG2 possesses a flat hydrophobic pocket, and ABCC1 exhibits amphiphilic properties in its pocket. Prior studies have underscored the significance of flat ring structures and hydrophobic features in inhibiting ABCG2 [143,144]. Until now, despite the determination of high-resolution structures of ABC transporters in various drug-binding conformations over the years, the notable advancements in structural biology have had limited impact on the design of improved inhibitors for multidrug resistance (MDR). However, as the number of resolved transporter structures continues to grow, computer-aided drug design, particularly with the integration of AI algorithms, holds the potential to accelerate the development of a new generation of clinically applicable inhibitors capable of reversing MDR in cancer therapy.

## 6. Concluding Remarks

In conclusion, this comprehensive review has provided an in-depth analysis of the structural aspects of MDR ABC transporters in humans, encompassing ABCB1, ABCG2, ABCC1, ABCC3, and ABCC4. Through the examination of high-resolution structural data, we have gained valuable insights into the intricate mechanisms underlying drug recognition, translocation, and inhibition in these essential transporters.

The elucidation of the central substrate-binding pockets in these transporters has highlighted their remarkable flexibility and plasticity, enabling them to accommodate a wide range of substrates, both hydrophobic and amphiphilic. This adaptability underscores their significance in multidrug resistance, making them formidable challenges in cancer therapy.

The “alternating access” model presented here has shed light on the general mechanism of drug translocation by MDR-related ABC transporters. This model underscores the pivotal role of large conformational changes involving transitions between inward-facing and outward-facing states, driven by ATP binding and hydrolysis. While variations exist among different members, this fundamental framework provides a unifying perspective on their transport mechanisms.

Our exploration of inhibition mechanisms has revealed two main strategies: small molecules/peptides and inhibiting antibodies. These inhibitors act by either sterically arresting the occluded conformation or locking the transporter in a specific state, thereby impeding drug efflux. These insights pave the way for the development of more effective inhibitors to combat multidrug resistance.

With the ever-expanding repository of high-resolution structural data, we anticipate that structure-based drug design will play a pivotal role in the development of novel inhibitors with improved specificity and affinity. The availability of these structures opens exciting avenues for the discovery of clinically applicable inhibitors to overcome multidrug resistance, thereby offering hope to cancer patients.

## Figures and Tables

**Figure 1 biomolecules-14-00231-f001:**
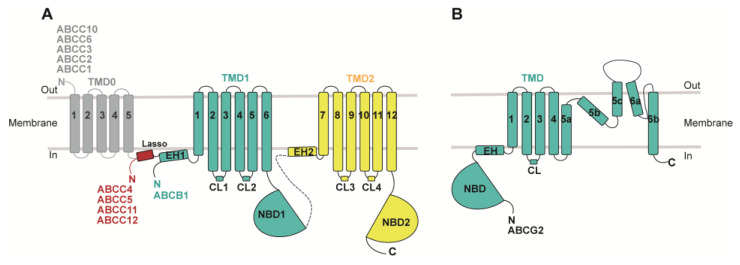
**Topology diagrams of MDR-related ABC transporters.** (**A**) ABCB1 and MRPs (ABCC1–6, ABCC10–12). (**B**) ABCG2. Regions not included in the models are represented by dashed lines. Lasso indicates the lasso motif; EH indicates the elbow helices.

**Figure 2 biomolecules-14-00231-f002:**
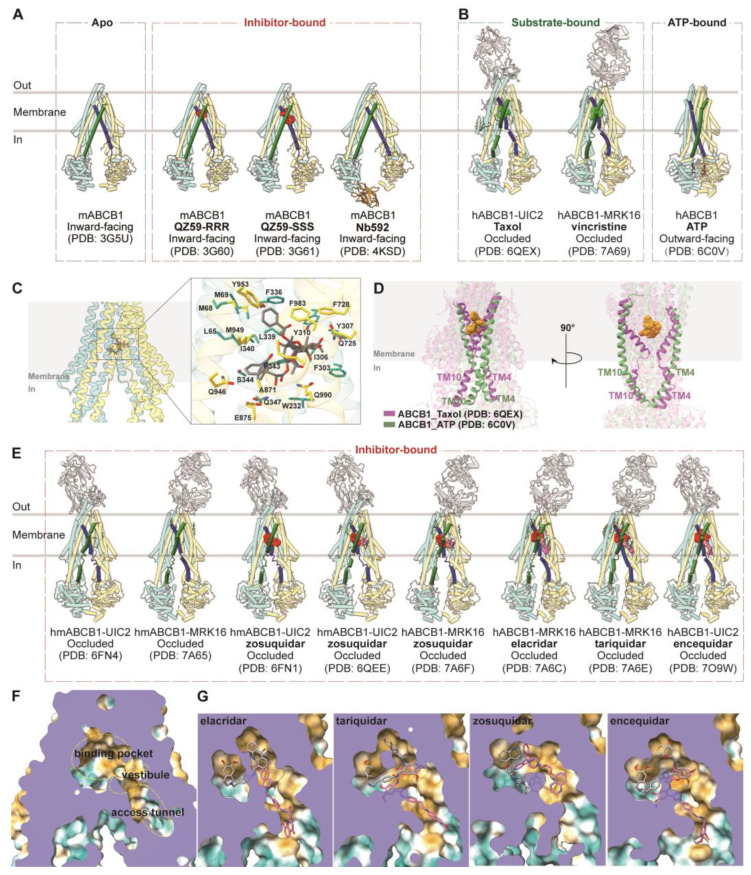
**Structures of ABCB1 and central binding cavity.** (**A**) Crystal structures of mABCB1 in apo (PDB code: 3G5U), cyclic peptide-inhibited (PDB codes: 3G60 and 3G61), and nanobody-inhibited (PDB code: 4KSD) conformations. TM4 and TM10 are colored dark blue and dark green, respectively. The cyclic peptides QZ59-RRR and QZ59-SSS are colored red. The nanobody Nb592 is colored gold. (**B**) Overview of human ABCB1 (hABCB1) and human-mouse ABCB1 (hmABCB1) structures in substrate-bound and nucleotide-bound states in cartoon representations. (**C**) Central binding cavity of ABCB1 bound with chemotherapy drug Taxol (PDB code: 6QEX). The Taxol molecule is shown as sticks. The right panel shows a zoomed-in view of Taxol binding. Side chains of residues within 5 Å are shown as sticks and indicated. (**D**) Superposition of ABCB1 shows a comparison of TM4 and TM10. ABCB1 bound with ATP (PDB code: 6C0V) is colored green, and ABCB1 bound with substrate Taxol (PDB code: 6QEX) is colored purple. Taxol molecules are shown as spheres and colored orange. (**E**) Overview of hABCB1 and hmABCB1 structures in inhibitor-bound states in cartoon representations. TM4 and TM10 are colored dark blue and dark green, respectively. (**F**) Surface representation of central binding cavity of ABCB1 (PDB code: 7A65). The binding pocket, vestibule, and access tunnel are indicated with dashed lines. (**G**) Close-up views of ABCB1 binding cavity in surface representations bound with inhibitor elacridar (PDB code: 7A6C), tariquidar (PDB code: 7A6E), zosuquidar (PDB code: 7A6F), and encequidar (PDB code: 7O9W). The surface is colored by hydrophobicity (yellow: hydrophobic; green: hydrophilic). The substrate and inhibitor molecules are shown as sticks.

**Figure 3 biomolecules-14-00231-f003:**
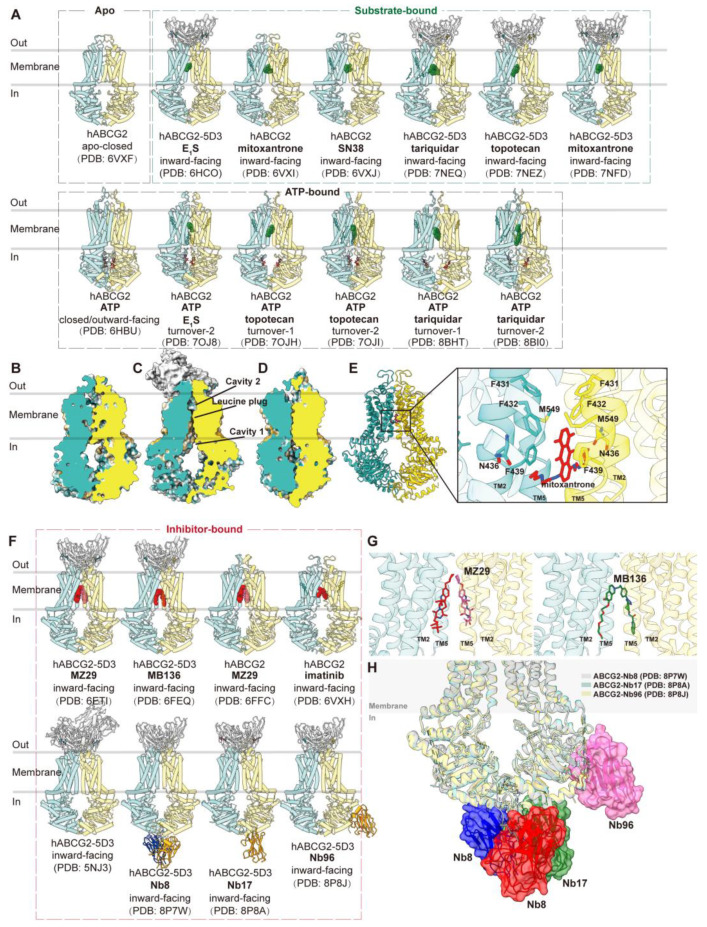
**Structures of ABCG2 and central binding cavity.** (**A**) Overview of human ABCG2 (hABCG2) structures in apo, substrate-bound, and nucleotide-bound states in cartoon representations. (**B**–**D**) Vertical slice-through surface representations of ABCG2 in three conformations: apo-closed (**B**, PDB code: 6VXF), inward-facing (**C**, PDB code: 5NJ3); outward-facing (**D**, PDB code: 6HBU). Cavity 1, cavity 2, and leucine plug are indicated in (**C**). (**E**) Cartoon representation of ABCG2 bound with chemotherapy drug mitoxantrone (PDB code: 6VXI). The right panel shows the zoomed-in view of mitoxantrone (red sticks) in the binding site. The interacting residues are indicated, and the side chains are shown as sticks. (**F**) Overview of hABCG2 in inhibitor-bound states in cartoon representations. (**G**) Close-up views of two molecules of inhibitor MZ29 (red and pink sticks, PDB code: 6ETI) and a single molecule of inhibitor MB136 (green sticks, PDB code: 6FEQ) in the binding site. Two monomers of ABCG2 are colored blue and yellow. (**H**) Superposition of ABCG2 structures bound with inhibitory nanobody Nb8 (red and blue, PDB code: 8P7W), Nb17 (green, PDB code: 8P8A), and Nb96 (pink, PDB code: 8P8J).

**Figure 4 biomolecules-14-00231-f004:**
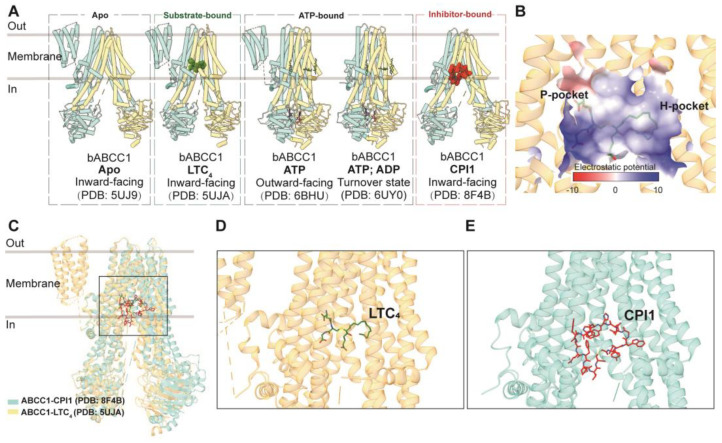
**Structures of ABCC1 and central binding cavity.** (**A**) Overview of bovine ABCC1 (bABCC1) structures in apo, substrate/inhibitor-bound, and nucleotide-bound states in cartoon representations. (**B**) The central binding pocket of ABCC1 showing electrostatic potential (PDB 5UJA). (**C**) Superposition of substrate-bound (LTC_4_, PDB code: 5UJA) and inhibitor-bound (CPI1, PDB code: 8F4B) bABCC1 structures colored in yellow and blue, respectively. (**D**) Zoomed-in view of LTC_4_ (green sticks) in the binding site. (**E**) Zoomed-in view of CPI1 (red sticks) in the binding site.

**Figure 5 biomolecules-14-00231-f005:**
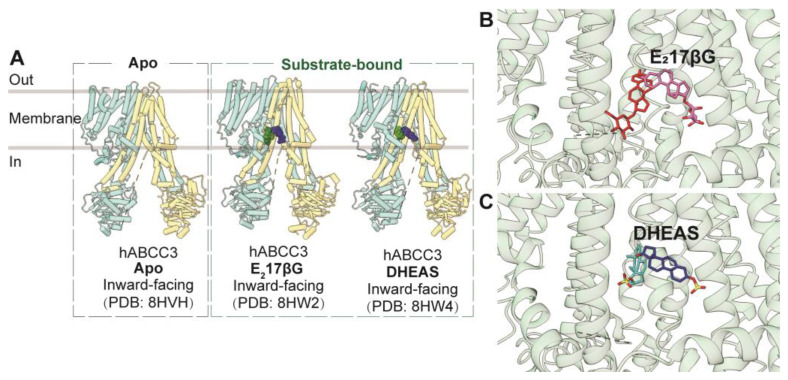
**Structures of ABCC3 and central binding cavity.** (**A**) Overview of human ABCC3 (hABCC3) structures in apo and substrate-bound states in cartoon representations. (**B**,**C**) Close-up views of a pair of E_2_17βG ((**B**), red and pink sticks, PDB code: 8HW2) and DHEAS ((**C**), cyan and dark blue sticks, PDB code: 8HW4) molecules in the central binding pocket of ABCC3 in cartoon representation colored in green.

**Figure 6 biomolecules-14-00231-f006:**
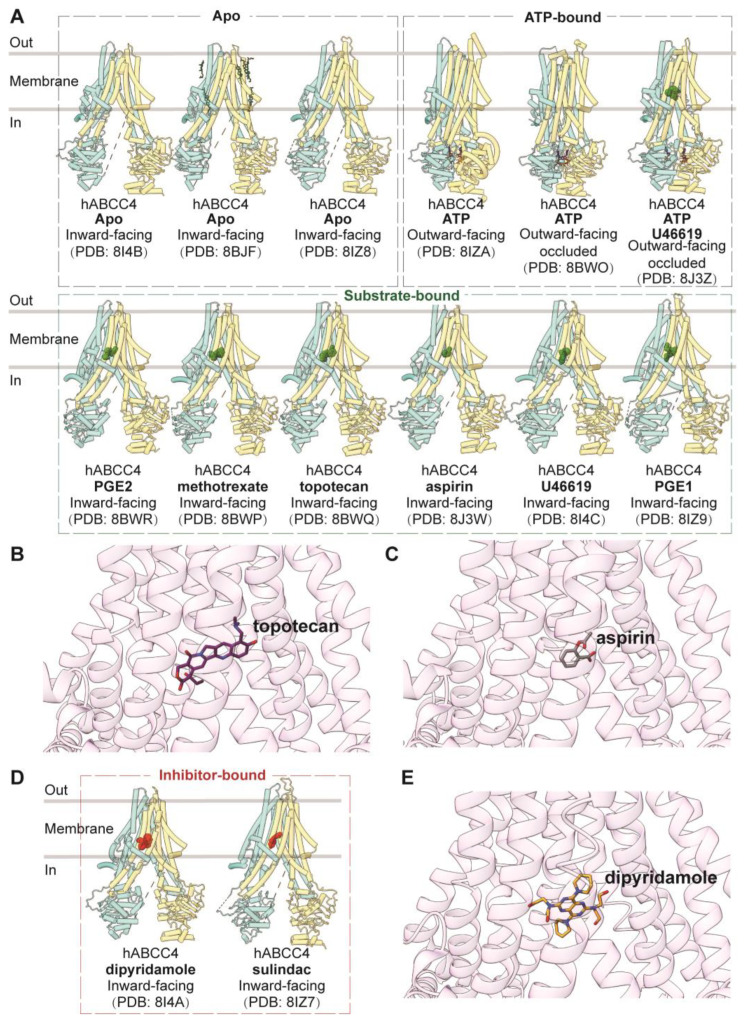
**Structures of ABCC4 and central binding cavity.** (**A**) Overview of human ABCC4 (hABCC4) structures in apo, nucleotide- and substrate-bound states in cartoon representations. (**B**,**C**) Close-up views of topotecan (purple sticks, PDB code: 8BWQ) and aspirin (grey sticks, PDB code: 8J3W) in the central binding pocket of ABCC4 in cartoon representation colored in pink. (**D**) Overview of hABCC4 structures in inhibitor-bound states in cartoon representations. (**E**) Close-up view of inhibitor dipyridamole (yellow sticks, PDB code: 8I4A) in the central binding pocket of ABCC4.

**Figure 7 biomolecules-14-00231-f007:**
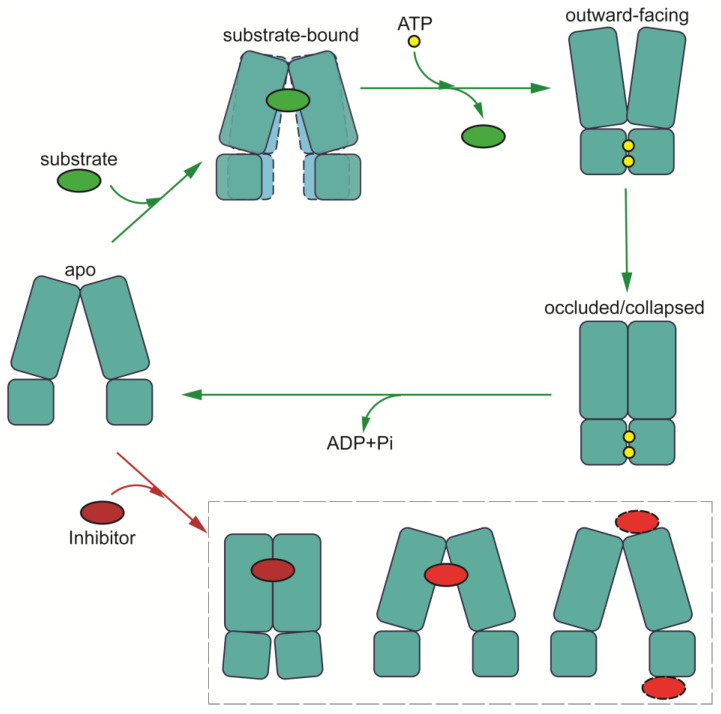
**Schematic of proposed transport cycle in the presence of substrate (green circles) and inhibitor (red circles).** Major conformational states are indicated: apo conformation; substrate-bound conformations; outward-facing conformation; and outward occluded/collapsed post-translocation state. Inhibitor-bound states are shown in the lower part of the diagram (occluded conformation for ABCB1; inward-facing conformation for MRPs; and inhibitory nanobodies bound at extracellular side of TMD or NBDs for allosteric inhibition of conformation). ATPs are indicated by yellow circles.

**Figure 8 biomolecules-14-00231-f008:**
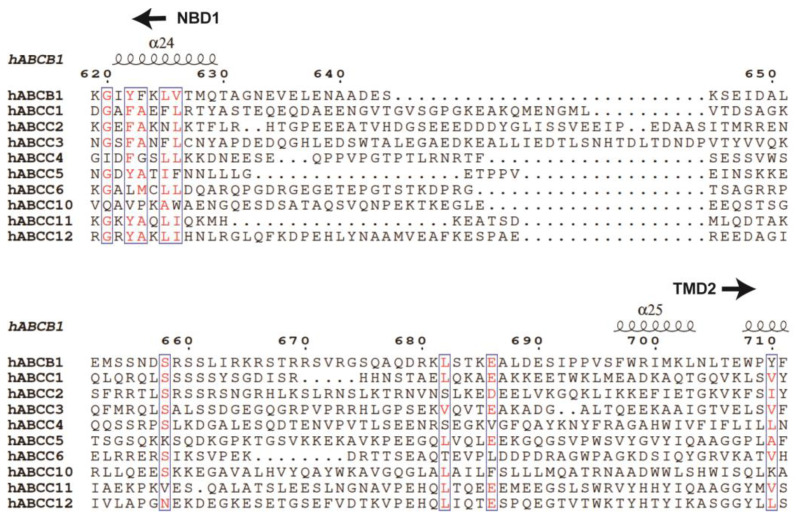
**Multiple sequence alignment of human ABCB1 (hABCB1) and MRPs (hABCC1–ABCC6, hABCC10–ABCC12) of the central linker region.** Representative secondary structural elements of hABCB1 are indicated above the sequences. Highly conserved residues are colored red.

**Table 1 biomolecules-14-00231-t001:** **Summary of active residues, substrates, inhibitors, function, and mechanism of human MDR-related ABC transporters**. * The residues highlighted with blue color represent active residues that also function in the binding of inhibitors.

Name (Alternate)	Active Sites in Central Cavity	Substrates	Inhibitors	Function and Mechanism
ABCB1 (P-gp, MDR1)	Y310 *, I340, F343, S344, Q347, Q725, Q946, Y953, F983, M986, A987, Q990 [46,47]	A variety of chemotherapy drugs (Taxol, vincristine, doxorubicin)	Cyclic peptide (QZ59-RRR, QZ59-SSS, cyclosporine A) Small molecules inhibitor (zosuquidar, elacridar, tariquidar, encequiar, verapamil) Nanobody (Nb592)	MDR; transports poly-specificity substrates; globular hydrophobic pocket primarily suitable for hydrophobic or weakly amphipathic compounds
ABCG2 (BCRP)	L405, F432, T435, N436, F439, S440, V442, T542, V546, M549 [48,49,50,51]	Chemotherapy drugs (topotecan, SN-38, mitoxantrone, doxorubicin) Endogenous substrates (sulfate conjugates of steroids, uric acid, and porphyrins, e.g., E_1_S) Exogenous cytotoxic compounds	FTC, Ko143, MZ29 Tariquidar, MB136 Tyrosine kinase inhibitors (TKIs) Antibody and nanobody (5D3, Nb8, Nb17, Nb96)	MDR; transports a broad spectrum of substrates; the deep/slit-like cavity preference for flat molecules
ABCC1 (MRP1)	K332, H335, L381, F385, Y440, T550, W553, F594, M1092, R1196, Y1242, N1244, W1245, R1248 [52,53]	Physiological substrates, xenobiotic compounds (leukotriene C4-LTC_4_, E_2_17βG) GSH and GSH-conjugated molecules	Indomethacin, verapamil, and its derivatives Macrocyclic peptide inhibitor (CPI1)	MDR; transports a wide range of substrates; employs large bipartite pockets to recognize amphiphilic substrates
ABCC3 (MRP3)	K318, Y371, F375, F426, L429, T535, W539, L580, M584, M1089, Y1188, R1193, F1238, N1241, W1242, R1245, M1246 [54]	Endogenous metabolites (bilirubin glucuronides, bile acids, and steroid hormones, like E_2_17βG, DHEAS) Anticancer drugs (etoposide, teniposide, methotrexate, and vincristine)	Cyclosporine A, MK-571	MDR; transports diverse endogenous metabolites and a limited number of anticancer drugs; the binding pockets comprise a substantial hydrophobic surface and two polar patches
ABCC4 (MRP4)	F211, F324, L363, L367, F368, R375, R946, Q994, W995, R998 [55,56]	Physiological substrates (cyclic nucleotides, steroid conjugates, folate, and prostaglandins (PGE1 and PGE2)). Xenobiotics (antibiotics, antiviral agents, anticancer drugs, aspirin, U46619)	Dipyridamole and sulindac	MDR; transports physiological substrates and xenobiotics; possesses a hydrophobic pocket (H-pocket) and positively charged pocket (P-pocket)

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
