# Peer review of "Structural View of Cryo-Electron Microscopy-Determined ATP-Binding Cassette Transporters in Human Multidrug Resistance"

_biomolecules, 2024, doi:10.3390/biom14020231_

Round 1

Reviewer 1 Report

Comments and Suggestions for Authors

Dear Authors,

Thank you for your valuable contribution. The review is well written and it will have a way to huge readers but if at all it is simplified.

To be honest, this review is just a collection of facts which needs easier representation. Make suitable tables like aa residues for active sites, category of drugs, mechanism, function.

Apart from this few suggestions,

Add ref in line 98.

"These structures reveal a pair of inhibitor 145 molecules occupying the central cavity, where substrate drug molecules"..it will be beneficial if residue numbers are highlighted.

Line 148 why suddenly a pmid appears instead of a citation.

Several more citations are missing. Please revise carefully.

Author Response

Reviewer 1:

Dear Authors,

Thank you for your valuable contribution. The review is well written and it will have a way to huge readers but if at all it is simplified.

Thank you for your positive feedback and valuable insights, particularly regarding the easier representation using tables. We have revised our manuscript according to your suggestions. This makes our manuscript more readable.

To be honest, this review is just a collection of facts which needs easier representation. Make suitable tables like aa residues for active sites, category of drugs, mechanism, function.

Thank you for your feedback. The table1 was added in a separate page (L117-120) in revised manuscript to summarise aa residues for active sites, category of drugs, mechanism, and function of ABCB1, ABCG2 and ABCC family proteins.

Apart from this few suggestions,

Add ref in line 98.

Thank you for pointing this out. A reference [32] has been added in revised text in L102.

"These structures reveal a pair of inhibitor 145 molecules occupying the central cavity, where substrate drug molecules”. It will be beneficial if residue numbers are highlighted.

Thank you for your suggestion. We analysed the substrate drug and inhibitor bound ABCB1 structures. Several key residues comprising the central cavity and participating substate/inhibitor binding were proposed in revised text L174-L176. A systemic comparation of substrate and inhibitor-bound residues were also listed in Table 1.

Line 148 why suddenly a pmid appears instead of a citation.

Thank you for pointing this out. It was cited as [58] in revised manuscript L178.

Several more citations are missing. Please revise carefully.

Thank you for your feedback. We have carefully checked our manuscript and added related citations in revised manuscript in L39, L43, L93, L291, L293 and L332.

Reviewer 2 Report

Comments and Suggestions for Authors

The manuscript presents a comprehensive overview of recent cryo-EM structures elucidating ABC transporters involved in multidrug resistance in cancer, including ABCB1, ABCG2, ABCC1, ABCC3 and ABCC4. The overall organization of text and figures is good, but I propose some essential revisions to enhance the depth and coherence of the review.

Major comments:

1.     The abstract mentions the role of ABC transporters in neurological disorders, which is not explored in the main text. This disconnect should be addressed for a more cohesive narrative.

2.      While the focus of this review is on cryo-EM structures, acknowledging the contributions of other biophysical studies, especially X-ray crystallography, in earlier years is crucial. This broader perspective will enrich the understanding of mammalian ABC transporter structures.

3.      Highlighting previously reported inhibitor-bound mouse P-glycoprotein structures and comparing them with the cryo-EM structures in the review could provide additional insights into drug binding pockets and inhibition mechanisms of ABCB1.

4.      In Figure 7, the proposal of an occluded conformation for ABCB1 induced by substrate or inhibitor binding should be reconsidered, given that similar conformation was also observed in apo conditions. Additionally, the use of detergent micelles or nanodiscs in different studies might have influenced the experimentally determined conformations. Given the large bodies of structural and conformational studies on P-glycoprotein in the literature, it might be premature to conclude that substrate or inhibitor binding would induce an occluded conformation for this particular transporter. 

5.      Acknowledge that high-resolution structures of some ABC transporters have been determined for many years. However, this remarkable progress in structural biology has not yielded the design of better MDR inhibitors.

6.      The manuscript discussed the different inhibition mechanisms of ABC transporters. Can the authors further discuss any advantages and disadvantages of antibody or nanobody inhibitors compared to small molecules? Additionally, nanobody-bound P-glycoprotein structures have been reported in the literature. It would be interesting to draw comparisons with ABCG2 for a more comprehensive understanding of the inhibition mechanisms.

Minor comments:

1.      Figure 1: Include clarifications for structural motifs, such as 'Lasso' and 'EH' abbreviations, either in the legend or main text.

2.      Line 103: Replace "biochemically unrelated substrates" with "chemically unrelated substrates"?

3.      In Figure 3A, describe the first structure on the second row (PDB: 6HBU) as 'closed/outward-facing' to align with the terminology used in the main text.

4.      Line 484: Omit the word 'relatively' or use the phrase 'relatively small-size molecules or peptides' for improved clarity.

Comments on the Quality of English Language

The writing is overall ok, but can be improved with the help of a professional editor. 

Author Response

The manuscript presents a comprehensive overview of recent cryo-EM structures elucidating ABC transporters involved in multidrug resistance in cancer, including ABCB1, ABCG2, ABCC1, ABCC3 and ABCC4. The overall organization of text and figures is good, but I propose some essential revisions to enhance the depth and coherence of the review.

We are sincerely grateful for your positive feedback. The time and effort you dedicated to review our paper, as well as your recognition is highly appreciated. We have carefully revised the manuscript and incorporated detailed information. This has significantly improved quality and depth of our review.

Major comments:

1. The abstract mentions the role of ABC transporters in neurological disorders, which is not explored in the main text. This disconnect should be addressed for a more cohesive narrative.

Thank you for your feedback. We had mentioned the ABC transporters and neurological disorders in the revised text in L333-334. We further explored the relationship between ABC transporters and neurological disorders in revised text L111-116, L229-233 and L335-340 to avoid the disconnection.

2. While the focus of this review is on cryo-EM structures, acknowledging the contributions of other biophysical studies, especially X-ray crystallography, in earlier years is crucial. This broader perspective will enrich the understanding of mammalian ABC transporter structures.

Thank you for your valuable suggestion. We have incorporated the first crystal structure of ABC transporter BtuCD in 2002 that frames the basic ABC architecture and mechanism in revised text L89- 91, and the first crystal structures of ABCB1 in 2009 in revised text L123-125 to show our respects. The crystal structures of ABCB1, including apo, cyclic peptide bound and nanobody inhibited structures were also presented in Figure 2A and discussed in revised text L123-134. We have also included the comparation between inhibitor-bound crystal and cryo-EM structures refer to the comment #3 in revised text L166-169 and L174-176. We carefully checked structures of ABCG2, ABCC1, ABCC3 and ABCC4, but found their first full structures are all determined by cryo-EM.

3. Highlighting previously reported inhibitor-bound mouse P-glycoprotein structures and comparing them with the cryo-EM structures in the review could provide additional insights into drug binding pockets and inhibition mechanisms of ABCB1.

Thank you for your feedback. Refer to reply of comment #2. The structures were presented in Figure 2A, the drug binding pockets and inhibition mechanism were discussed.

4. In Figure 7, the proposal of an occluded conformation for ABCB1 induced by substrate or inhibitor binding should be reconsidered, given that similar conformation was also observed in apo conditions. Additionally, the use of detergent micelles or nanodiscs in different studies might have influenced the experimentally determined conformations. Given the large bodies of structural and conformational studies on P-glycoprotein in the literature, it might be premature to conclude that substrate or inhibitor binding would induce an occluded conformation for this particular transporter.

Thank you for your feedback. We have reconsidered the occluded conformation for ABCB1 and removed the related occluded cartoon for ABCB1 in Figure7 and revised figure7 legend (L561-564). We also revised our text in L545 to eliminate possible misunderstanding. We agree with that different micelles or nanodisc in different studies might have influenced the experimentally results. While the four structures (inhibitory antibody bound, with or without drug) determined by the Locher lab were reconstituted by LMNG/CHS (6QEX), amphipol (A8-34, 6FN4), nanodiscs (7A65) and nanodiscs (7A69), respectively. They all showed similar occluded conformations, so it may not be the case here.

5. Acknowledge that high-resolution structures of some ABC transporters have been determined for many years. However, this remarkable progress in structural biology has not yielded the design of better MDR inhibitors.

Thank you for your suggestion. We added the conclusion in revised manuscript L715-718.

6. The manuscript discussed the different inhibition mechanisms of ABC transporters. Can the authors further discuss any advantages and disadvantages of antibody or nanobody inhibitors compared to small molecules? Additionally, nanobody-bound P-glycoprotein structures have been reported in the literature. It would be interesting to draw comparisons with ABCG2 for a more comprehensive understanding of the inhibition mechanisms.

Sure. We have addressed advantages and disadvantages of antibody or nanobody inhibitors and small-size molecules in revised text L587-L592. We found only one nanobody-bound P-glycoprotein structure solved by crystallography at a resolution of 4.1AÌŠ published in 2013. We include this structure in Figure 2A (L184) and the comparisons between ABCB1 and ABCG2 bound to nanobodies or antibodies were presented in revised text L597-604, L608-614 and L617-622.

Minor comments:

1. Figure 1: Include clarifications for structural motifs, such as 'Lasso' and 'EH' abbreviations, either in the legend or main text.

Thank you for pointing this out. We have included their clarifications: Lasso, the lasso motif. EH, the elbow helices. in the revised Figure1 legend (L84-86).

2. Line 103: Replace "biochemically unrelated substrates" with "chemically unrelated substrates"?

Thank you for pointing this out. We have revised it as ‘chemically unrelated substrates’ in revised text L107.

3. In Figure 3A, describe the first structure on the second row (PDB: 6HBU) as 'closed/outward- facing' to align with the terminology used in the main text.

Thank you for pointing this out. We have replaced the text in Figure3A with ‘closed/outward-facing’ in revised manuscript L304.

4. Line 484: Omit the word 'relatively' or use the phrase 'relatively small-size molecules or peptides' for improved clarity.

Thank you for pointing this out. We have rephrased with ‘relatively small-size molecules or peptides’ in revised text L580.

Reviewer 3 Report

Comments and Suggestions for Authors

The manuscript has provided a comprehensive review of MDR-related human ABC transporters from the structural aspect and an insightful discussion of the MDR reversal strategies based on the protein and drug structures. 

Some more information is suggested to include in this review article from my view:

1. For ABCB1, some recent structural studies used genetic modification to discover the role of other transmembrane helixes (such as TMH 1, 6, 7, 12) and some key residues in drug binding, protein conformational change, and transport. Some of these studies discussed the potential of DNA therapies to change ABCB1 structure and transport direction to overcome MDR, though still in the experimental phase.  These studies should also be reviewed and discussed.

2. The authors should review and discuss, from the aspects of structure and the affected transport function, the alteration of ABCG2 substrate/inhibitor specificity by mutation at known hotspots, such as residue 482.

3. In the sections of ABCCs, the authors should discuss the glutathione (GSH) co-transport and GSH-stimulated transport mechanism. GSH plays a key role in the transport activity of certain ABCCs, particularly ABCC1. Some ABCC1 inhibitors reverse MDR by triggering ligand-stimulated GSH transport of ABCC1. The related information should be included in this review article.

4. ABCB6 is also one of the MDR-related ABC transporters in the treatment of certain types of cancer. It has been reported that ABCB6 is associated with resistance to anti-cancer drugs such as SN-38, vincristine, and metal-containing chemotherapies like platinum compounds and arsenic agents. Additionally, the cryo-EM structure of human ABCB6 has been determined (PDB: 7DNY, 7DNZ, 7D7N, 7D7R,8K7B, 8K7C, 8FWK, etc.).  GSH also plays a role in ABCB6-related MDR. A section to review related information on ABCB6 is highly recommended.

5. Currently only around one-third of the references are publications within the past 10 years. It is reasonable to cite some important old articles. However, a review discussing more recent findings is expected. The authors may consider adding the above-suggested information by reviewing more research discoveries from recent years. 

Author Response

The manuscript has provided a comprehensive review of MDR-related human ABC transporters from the structural aspect and an insightful discussion of the MDR reversal strategies based on the protein and drug structures.

Thank you for your positive feedback and constructive suggestions. We added some systematic discussion and new citations as your suggestions. These new contents provide wide insights and strengthen our manuscript.

Some more information is suggested to include in this review article from my view:

1. For ABCB1, some recent structural studies used genetic modification to discover the role of other transmembrane helixes (such as TMH 1, 6, 7, 12) and some key residues in drug binding, protein conformational change, and transport. Some of these studies discussed the potential of DNA therapies to change ABCB1 structure and transport direction to overcome MDR, though still in the experimental phase. These studies should also be reviewed and discussed.

Thank you for your valuable perspective. We have reviewed and discussed new strategies such as genetic modifications were carried out to study the roles of ABCB1 transmembrane helixes (such as TM1/7 and TM6/12) and discuss the potential DNA therapies to change substrate specificity and transport direction to overcome MDR in revised text in L204-215.

2. The authors should review and discuss, from the aspects of structure and the affected transport function, the alteration of ABCG2 substrate/inhibitor specificity by mutation at known hotspots, such as residue 482.

Thank you for your suggestion. The most hot-shop residue Arg482 of ABCG2 locates outside of the substrate/inhibitor binding cavity/slit-like cavity 1 but interacts with TM2 (the cavity is formed by TM2 and TM5) through hydrogen bonds. Its mutation may reshape the cavity and thus changes substrate/inhibitor specificity. Other ABCG2 mutations were also discussed after residue 482. We have revised our manuscript to discuss the hot spot sites in revised text L265-279.

3. In the sections of ABCCs, the authors should discuss the glutathione (GSH) co-transport and GSH- stimulated transport mechanism. GSH plays a key role in the transport activity of certain ABCCs, particularly ABCC1. Some ABCC1 inhibitors reverse MDR by triggering ligand-stimulated GSH transport of ABCC1. The related information should be included in this review article.

Thank you for your feedback. We accepted the reviewer’s advice and have discussed the GSH and GSH conjugated molecules transport of ABCC1 in a new section 2.3.1.3 (in revised text L411-434). The ABCB6 bound to GSH and GSH conjugated hemin is also included in this section after ABCC1 (in revised text L426-434).

4. ABCB6 is also one of the MDR-related ABC transporters in the treatment of certain types of cancer. It has been reported that ABCB6 is associated with resistance to anti-cancer drugs such as SN-38, vincristine, and metal-containing chemotherapies like platinum compounds and arsenic agents. Additionally, the cryo-EM structure of human ABCB6 has been determined (PDB: 7DNY, 7DNZ, 7D7N, 7D7R,8K7B, 8K7C, 8FWK, etc.). GSH also plays a role in ABCB6-related MDR. A section to review related information on ABCB6 is highly recommended.

Thank you for your feedback. We have noticed the mitochondrial out membrane transporter ABCB6 and its role associated with resistance to anti-cancer drugs. While ABCB6 commonly functions in transporting heme or its precursors in mitochondria, and GSH plays a critical role in ABCB6 substrate translocation and toxic mental resistance. Until now, there were no cryo-EM or crystal structures of ABCB6 that bind cytotoxic drugs. Additionally, the cryo-EM structures of ABCB6 were still under study in our lab and we already have some new clues. Hence, we just discuss the GSH and GSH conjugated molecules transport of ABCB6 together with ABCC1.

5. Currently only around one-third of the references are publications within the past 10 years. It is reasonable to cite some important old articles. However, a review discussing more recent findings is expected. The authors may consider adding the above-suggested information by reviewing more research discoveries from recent years.

Thank you for your suggestion. We have added some new citations of your mentioned information and somewhere else. Two citations [67] and [68] were added in revised manuscript for potential DNA therapies to change ABCB1 structure and transport direction to overcome MDR in revised text L207. Citations [77-81] were added in part of ABCG2 hot spots in revised text L267-279. The citations [106- 110] were added in section of GSH transport of ABCC1 in revised text L411-434. Citations [88] and [89] about ABCC1 and neurological disorders were added in revised text L339-340.

Round 2

Reviewer 1 Report

Comments and Suggestions for Authors

Dear Authors,

The improvements have been incorporated and its so good now. Congratulations on your MS.

Reviewer 2 Report

Comments and Suggestions for Authors

The authors have addressed my major concerns in this revised manuscript. I recommend the acceptance of this version for publication.